# Systematic Evaluation of Antigenic Stimulation in Chronic Lymphocytic Leukemia: Humoral Immunity as Biomarkers for Disease Evolution

**DOI:** 10.3390/cancers15030891

**Published:** 2023-01-31

**Authors:** Alicia Landeira-Viñuela, Miguel Alcoceba-Sanchez, Almudena Navarro-Bailón, Carlota Arias-Hidalgo, Pablo Juanes-Velasco, José Manuel Sánchez-Santos, Quentin Lecrevisse, Carlos Eduardo Pedreira, Marina L. García-Vaquero, Ángela-Patricia Hernández, Enrique Montalvillo, Rafael Góngora, Javier De las Rivas, Marcos González-Díaz, Alberto Orfao, Manuel Fuentes

**Affiliations:** 1Department of Medicine and General Service of Cytometry, CIBERONC-CB16/12/00400, Cancer Research Centre-IBMCC, CSIC-USAL, IBSAL, Campus Miguel de Unamuno s/n, University of Salamanca-CSIC, 37008 Salamanca, Spain; 2Department of Hematology, Center Research-Centre IBMCC (CSIC-USAL, IBSAL), University Hospital of Salamanca, CIBERONC-CB16/12/00233, 37007 Salamanca, Spain; 3Statistics Department, University of Salamanca, 37008 Salamanca, Spain; 4Systems and Computing Department (COPPE-PESC), Universidade Federal do Rio de Janeiro (UFRJ), Rio de Janeiro 21941-914, Brazil; 5Organic Chemistry Section, Department of Pharmaceutical Sciences, Faculty of Pharmacy, University of Salamanca, Campus Miguel de Unamuno s/n, 37008 Salamanca, Spain; 6Bioinformatics and Functional Genomics Group, Cancer Research Center (CiC-IBMCC, CSIC/USAL), Consejo Superior de Investigaciones Científicas (CSIC) and University of Salamanca (USAL), 37008 Salamanca, Spain; 7Proteomics Unit, Cancer Research Centre-IBMCC, IBSAL, Campus Miguel de Unamuno s/n, University of Salamanca-CSIC, 37008 Salamanca, Spain

**Keywords:** chronic lymphocytic leukemia, antimicrobial antibodies, autoantibodies, protein microarrays

## Abstract

**Simple Summary:**

Clonal B cell expansion in chronic lymphocytic leukemia may be triggered by persistent antigenic stimulation. Therefore, studying the dynamics of the humoral response of these patients provide information about serological immunoglobulin levels and identifies which antigens compromise the immune response. Hence, this study is a new point of view that could provide additional information useful in patient stratification and also selection of targeted therapy.

**Abstract:**

Chronic lymphocytic leukemia (CLL) is the most common leukemia in the Western world. Studies of CLL antibody reactivity have shown differential targets to autoantigens and antimicrobial molecular motifs that support the current hypothesis of CLL pathogenesis. Methods: In this study, we conducted a quantitative serum analysis of 7 immunoglobulins in CLL and monoclonal B-cell lymphocytosis (MBL) patients (bead-suspension protein arrays) and a serological profile (IgG and IgM) study of autoantibodies and antimicrobial antigens (protein microarrays). Results: Significant differences in the IgA levels were observed according to disease progression and evolution as well as significant alterations in IgG1 according to IGHV mutational status. More representative IgG autoantibodies in the cohort were against nonmutagenic proteins and IgM autoantibodies were against vesicle proteins. Antimicrobial IgG and IgM were detected against microbes associated with respiratory tract infections. Conclusions: Quantitative differences in immunoglobulin serum levels could be potential biomarkers for disease progression. In the top 5 tumoral antigens, we detected autoantibodies (IgM and IgG) against proteins related to cell homeostasis and metabolism in the studied cohort. The top 5 microbial antigens were associated with respiratory and gastrointestinal infections; moreover, the subsets with better prognostics were characterized by a reactivation of Cytomegalovirus. The viral humoral response could be a potential prognosis biomarker for disease progression.

## 1. Introduction

Chronic lymphocytic leukemia (CLL) is one of the most common leukemias present in the Western adult population. The clinical manifestation of this disease is an accumulation of clonal B lymphocytes (B-CLLs) in peripheral blood (PB), bone marrow, and lymphoid organs. These B-CLLs are CD5^+^, CD19^+^, CD23^+^; there are also low levels of CD21, CD22, CD79, and FcRIIb and high levels of CD23, CD25, CD69, CD71, and the surface immunoglobulins (Ig) IgM and IgD that are accompanied by genetic mutations throughout disease development and progression [1,2,3,4,5]. There is a previous stage called monoclonal B-cell lymphocytosis (MBL), which is characterized by the presence of B-CLLs in PB and the absence of clinical manifestations [6,7]. In general, CLL is a highly heterogeneous disease from clinical and biological points of view. At the clinical level, there are asymptomatic patients whose disease could remain stable for years without any therapeutically intervention; in other cases, they present an aggressive form of the disease that requires early treatment and presents a short survival. From the biological point of view, there are the cytogenetics alterations (del (13q), del (11q), trisomy 12, and del (17p)), abnormalities at the protein and/or mRNA expression levels, single-nucleotide polymorphisms (SNPs), and point mutations (presence/absence) in the genes of the variable region of the immunoglobulin heavy chain (IGHV) [3,6,7,8,9,10].

Another characteristic of the CLL is the alteration of the immune system in terms of both the innate and adaptive immune responses. These alterations affect immune surveillance, which allows tumor progression and therefore affects the course of the disease [5,11,12]. This is due to the interconnection and intercommunication of the B-CLLs with other cells of the immune system. The importance of the tumor microenvironment (TME) is well known in terms of the cellular recruitment and alteration of nontumor cells (such as the expansion of T lymphocytes and exhausted NK cells) and/or the decrease in circulating normal B cells; thus, the TME increases a patient’s susceptibility to infections, autoimmune phenomena, and/or the development of secondary malignancies [7,12,13,14].

B cells are responsible for the humoral response (because they produce antibodies—Abs-) and also act as antigen-presenting cells (APCs). The selection, development, proliferation, and survival of B cells are determined by the activation of BCR signaling pathways in response to antigen stimulation. BCR is a membrane Ig that is capable of detecting microbial antigens (mAgs) and also autoantigens (aAgs), which are directly involved in triggering processes related to B-CLL survival and proliferation (whether pathological or not) [3,15].

Currently, the IGHV mutational status is one of the biomarkers that allows the stratification of CLL patients and is stable from the onset and development of the disease. It is well known that when IGHV is mutated (M-IGHV), patients present an indolent disease with an anergic and antiapoptotic response, while those CLL patients without IGHV mutations (U-IGHV) have an active and proliferative response that leads to a rapid evolution of the disease and consequently a short survival [15,16].

Therefore, an uncontrolled proliferation of B-CLLs facilitates the acquisition of genetic mutations over time; however, the inherent somatic hypermutation (Ig heavy chain or light chain) of B cells should also be noted. In fact, somatic hypermutation is responsible for producing isotype changes in the BCR. This process can only occur in germinal centers on activated naïve B cells and memory B lymphocytes before their differentiation to plasma cells (antibody (Ab)-secreting cells) [5,17]. A stererotyped BCR is presented in 30% of CLL patients that is based on the structural similarity of the complementary determining regions, thus forming a restricted set of Ig genes (IGHV1-69, IGHV3-7, IGHV3-21, and IGHV4-34 as the more featured stereotypes) [18]. In addition, these B-CLLs commonly present IgM in their membranes (Ig reactive to a wide variety of antigen epitopes); however, the autoimmunity developed by these CLL patients seems to be restricted to aAgs mostly expressed by blood cells (erythrocytes, platelets, and granulocytes) and mediated by IgG autoantibodies (aAbs) [5,19].

At the time of diagnosis in CLL patients, hypogammaglobulinemia is known, which mostly affects the levels of IgG, IgM, and IgA in the serum. This fact increases the risk of infections; therefore, the morbidity and mortality of CLL patients are also increased [7,20,21]. In addition to this fact, the disease stage and therapies are directly correlated with the risk of infection. Hence, studies have reported a wide range of infections in these patients that included from bacterial respiratory tract infections to erythema (such as those caused by *Streptococcus pneumoniae*, *Haemophilus influenzae*, *Staphylococcus aureus, Streptococcus pyogenes,* and *Escherichia coli*), viral reactivation (varicella-zoster, herpes simplex virus (HSV), cytomegalovirus (CMV), and Epstein–Barr), opportunistic infections (*Listeria, Aspergillus, Candida,* and *Cryptococcus*), parasites, and mycobacteria among many others [20,22,23,24].

Bearing all of this in mind, in this study, we quantitatively analyzed the Ig serum levels (IgM, IgA and IgG and their subclasses) at different disease stages (diagnosis and evolution). In addition, this CLL cohort was screened against a targeted panel of aAbs and antimicrobial Abs (mAbs) to decipher any relevant antigen (Ag) pattern in the disease progression and/or evolution.

## 2. Materials and Methods

### 2.1. Patient Cohort

A total of 67 PB samples (10 mL/case) were obtained in EDTA-coated tubes from 57 CLL diagnostic adults and 10 MBL diagnostic adults between May 2018 and October 2020 (29 females and 38 males; median age of 70 years ranging from 36 to 91 years) (Table 1). The diagnosis was made according to the National CLL Guidelines of the Spanish CLL Group (GELLC) based on the International Workshop on Chronic Lymphocytic Leukemia (iwCLL) [25], and the staging was performed according to Binet and Rai criteria [26,27]. Each individual signed an informed consent before participating in the study, which was approved by the local ethics committee of the University Hospital of Salamanca (HUS; Salamanca, Spain).

### 2.2. Serum Sample

Regarding the CLL samples, the PB samples (10 mL/case) were centrifuged immediately after collection at 800× *g* for 10 min (min) at room temperature (RT), and the plasma was stored in aliquots at −80 °C until further use.

Then, 1 mL of plasma from each patient was defrosted on ice, centrifuged for 5 min at 200× *g*, aliquoted at 100 µL/sample in 96-well polypropylene PCR microplates (Axygen, Somerville, MA, USA), and stored at −80 °C until further analysis.

For the reference serum sample (Ref.), we employed a standardized plasma from the National Institute of Standards and Technology (909c Frozen Human Serum, NIST, Gaithersburg, MD, USA). It was a pool of plasma from healthy individuals (age range: 25–80 years). This serum was diluted in PBS 1× Na^+^/K^+^. Aliquots of 30 µL were made and stored at −20 °C until further analysis.

### 2.3. Evaluation of Immunoglobulin Quantitative Levels

For all samples, we conducted a study of 7 isotypes of Ig (IgM, IgA, IgE, IgG1, IgG2, IgG3, and IgG4).

#### 2.3.1. Immunoglobulin Profiling

In this study, the 7 isotypes were simultaneously determined via Luminex technology (Luminex Inc, Brooklyn, NY, USA) in all serum samples. This determination was conducted using a Procarta Plex Human antibody isotyping kit (Invitrogen, ThermoFisher, Waltham, MA, USA) according to the manufacturer’s instructions. The MAGPIX^®^ reproducibility was evaluated with calibration and verification kits according to the manufacturer’s instructions (MPXIVD-CAL-K25 and MPXIVD-PVER-K25, respectively). The standard curve for each isotype was added in duplicate for each experiment. After acquisition, a five-parameter logistic (5PL) curve was obtained, and the standard recovery was calculated using the following equation:$$\frac{\left[\mathrm{Observed}\;\mathrm{concentration}\right]\;}{\left[\mathrm{Expected}\;\mathrm{concentration}\right]}\times 100$$

Values for samples above the recovery range (70% to 130%) were considered accurate as recommended by the manufacturer.

#### 2.3.2. Data Analysis of Immunoglobulin Profiles

The xPONENT^®^ software (version 4.2) for the Luminex instrument was used for data analysis. Data concentration was conducted using a 5PL curve-fitting algorithm, the equation for which was as follows:$$\;y=a+\frac{b-a}{{\left(1+{\left(\frac{x}{c}\right)}^{d}\right)}^{f}}\;$$
where *a*, *b*, *c, d,* and *f* are coefficients; *y* is the net median fluorescence intensity; and *x* is the concentration in pg/mL.

### 2.4. Evaluation of Autoantibody Serum Profiles

#### 2.4.1. Autoantibodies to Known Tumoral and Neoplasm Antigen Proteins

For the analysis of aAbs, we employed a commercial protein array containing 122 proteins (Appendix A) provided by GeneCopoeia^TM^ (Rockville, MD, USA). The serum screening was performed following the manufacturer’s instructions.

In this assay, the final working dilution (1:10 *v*/*v*) of serum samples was supplemented with DNase I (DNase I and associated buffer (cat. nos. M0303S and B0303S, respectively; New England BioLabs Inc., Ipswich, MA, USA), 0.1 M DTT, and nuclease-free water. Mix buffer was used as a negative control. All diluted serum samples and the control were incubated for 30 min with mild stirring at RT.

All the arrays were incubated with blocking buffer (PBST, 5% *p*/v BSA) (100 µL/well) for 30 min with shaking at RT. Washings (2×) were conducted with PBST (PBS 1× Na^2+^K^+^, 0.05% *v*/*v* Tween 20) for 5 min with shaking at RT.

The working dilutions of serum samples and the control were diluted 1:10 (*v*/*v*) in PBST (100 µL/well) and incubated for 1 h with mild stirring at RT. After that, arrays were washed (3×) at RT with mild stirring. The first and last washing were conducted with 100 µL/well of PBST. A second washing was conducted with 100 µL/well of blocking buffer.

Secondary Ab incubation was performed with 100 µL/subarray of a solution containing a cocktail of secondary Abs (Cy3-AffiniPure Donkey Anti-Human IgG, Cat. no. 709-165-149, Jackson ImmunoResearch, Ely, UK and Alexa Fluor 647-AffiniPure Goat Anti-Human IgM, cat. no. 109-605-043, Jackson ImmunoResearch, Ely, UK) at a final dilution 1:1000 (*v*/*v*) and incubated for 1 h with mild stirring at RT.

At least seven washings were performed with mild stirring for 5 min at RT. The first three washes were with 100 µL/well of PBST, the next two washes were with 45 mL/array of PBS 1× Na^2+^K^+^, and the last two washes were with 45 mL/array of nuclease-free water. Finally, the arrays were dried and scanned as described in Section 2.4.3.

#### 2.4.2. Profiling Antimicrobial Antibodies via Stimulation Persistent Antigenic Array

A stimulation-persistent antigenic array (SPA array) was designed and developed that contained 37 mAgs (Appendix A) among 10 array controls (positive and negative). The total array content was deposited via noncontact printer (Arrayjet Printer Marathon 1.4) using a 1:1 (*v*/*v*) dilution in Arrayjet printing buffer C and PBS 1× Na^2+^/K^+^.

The SPA array was designed with a slideout of 7 subarrays with 4788 total spots. Each subarray contained 5 serial dilutions of each mAg and IgG (range: 0.08 to $1\cdot 10^{-5}\;µg/µL$) among each Ab, and dye as controls (at 1 µg/mL) were also deposited. All of the 188 probes and controls were printed in triplicate on a chemically activated glass surface prepared according to a previously study by M. González-González et. al. in 2014 [28]. The microarrays were incubated in the dark; first for 2 days at RT and subsequently for 10 days at 37 °C. These dried arrays were packed and placed in dark storage at RT until further use.

All of the arrays were incubated with Blocker™ BSA 1X (cat. no. 37525, Thermo Fisher, Waltham, MA, USA) for 1 h with shaking and RT. Washings (3×) were conducted with distilled water for 5 min at RT with shaking.

Serum samples and the negative control (PBS 1× Na^2+/^K^+^) were diluted (1:10 *v*/*v*) in Blocker™ BSA 1×. Subsequently, 100 µL/subarray of this dilution was incubated overnight at 4 °C in an orbital shaker. After that, washings (3×) were conducted with PBST for 5 min at RT with mild stirring.

Secondary Ab incubation was conducted with 100 µL/well of cocktail of secondary Abs (Cy3-AffiniPure Donkey Anti-Human IgG; cat. no. 709-165-149, Jackson ImmunoResearch, Ely, UK; and Alexa Fluor 647-AffiniPure Goat Anti-Human IgM, cat. no. 109-605-043, Jackson ImmunoResearch, Ely, UK) at a final dilution 1:1000 (*v*/*v*) and incubated for 1 h with mild stirring at RT.

Seven washings were conducted with shaking for 5 min at RT. The first three washes were with 100 µL/well of PBST, the next two washes were with 45 mL/array of PBS 1× Na^2+^/K^+^, and the last two washes were with 45 mL/array of distilled water. Finally, the arrays were dried and scanned as described below.

#### 2.4.3. Image Acquisition

The scanning was conducted with a GenePix^®^ 4000B Microarray Scanner (Axon, Molecular Devices, San Jose, CA, USA). The parameters were set to quantify signal-intensity values for Cy3 (λ = 532 nm) and Alexa Fluor 647 (λ = 635 nm). The TIFF images obtained via the array scanning were analyzed using GenePix Pro 6.0 software (Molecular Devices, San Jose, CA, USA).

### 2.5. Data Analyses

For all analyses, the samples were classified according to the clinical characteristics of the patients (diagnosis: MBL and CLL; disease evolution: CLL stable/constant (c-CLL) and progression (p-CLL); line of treatment/therapy response: CLL group previous to first-line treatment (CLL-PFT)- and another group subsequent to first-line treatment (CLL-TFT); and IGHV mutational status: M-IGHV and U-IGHV) (Appendix A).

#### 2.5.1. Quantitative Determination of Immunoglobulin Level Profiles

Values under the detection limit were substituted with 10% of the lowest value for each analyte [29]. All of the results were reported in pg/mL and Log_10_ transformed for further analysis.

First, conventional statistics analyses were conducted using the software R v. 4.1 under RStudio v. 2022.02.0. The normality of the data was studied using the Shapiro–Wilk normality test. The homoscedasticity between categories of the factor variables was verified using the Fligner–Killeen test and Bartlett’s test. According to whether the variables were normal and homoscedastic/heteroscedastic, the search for significant differences between categories conducted using ANOVA (for more than two categories) and the *t*-test (two-by-two categories), respectively. When variables were non-normal and homoscedastic/heteroscedastic, the search for significant differences between categories was conducting using a Kruskal–Wallis test (for more than two categories) and the Wilcoxon–Mann–Whitney test (two-by-two categories), respectively. The post hoc tests used were Tukey’s, a *t*-test, or a Wilcoxon depending on the corresponding global test of significant differences. A significance level of 5% was set for all of the statistical tests.

Graphical representations and visualizations were achieved using spreadsheets, Infinicyt™ 2.0.5 (Cytognos SL, Salamanca, Spain), and GraphPad Prism Software v. 8.0.2 (GraphPad Software Inc., San Diego, CA, USA).

#### 2.5.2. Data Preprocessing and Dataset Analyses

The fluorescence signal retrieved from processed images was corrected by subtracting the background signal and transformed into a Z-score [30]. For each spot, the fluorescence signal was obtained according to following steps:(i)Signal normalization (S):$$S=\frac{\tilde{F_{\lambda }}-{\tilde{B}}_{\lambda }}{B\sigma_{\lambda }}\;$$
where $\tilde{F_{\lambda }}$ is the median signal intensity of the feature pixels, ${\tilde{B}}_{\lambda }$ is the median signal intensity of the background pixels, and $B\sigma_{\lambda }$ is the standard deviation for the background pixels at a selected wavelength (λ=532 nm or 635 nm) [31].(ii)Negative control ($S_{neg}$) signal intensity:$$S_{neg}=\frac{\sum S_{p}\;}{N}$$
where $S_{p}$ is the normalized signal for a selected spot (incubated with the negative control) and N is the number of subarrays (belonging to the same incubation).(iii)Spot signal intensity (I):$$I=S_{y}\;–S_{neg}$$
where $S_{y}$ is the normalized signal for the spot and $S_{neg}$ is the normalized signal for the corresponding negative control spot.(iv)Ag average signal intensity (${\bar{I}}_{m}$):$${\bar{I}}_{m}=\frac{\sum I_{y}}{Z}\;$$
where $I_{y}$ is the signal intensity at a selected spot and Z is the number of replicas.(v)Spot selection with respect to the Ref.: (a)Only Ags with a signal intensity > 0 were considered.(b)Selection of Ag-positive signals above compared to Ref.:$${\bar{I}}_{m}^{s}>{\bar{I}}_{m}^{r}+I_{m}^{r}\sigma$$
where ${\bar{I}}_{m}^{s}$ and ${\bar{I}}_{m}^{r}$ are the average signal intensity of the Ag for the sample and the Ref., respectively; and $I_{m}^{r}\sigma$ is the standard deviation signal intensity of the Ag for the Ref.(vi)Ag Z-score:$$Z\;score=\frac{{\bar{I}}_{m}^{s}}{{\bar{I}}_{m}^{r}+I_{m}^{r}\sigma }\;$$(vii)Number (*n*#) of Ag hits for each Ab isotype or clinical groups (hits).

In the case of the study of mAbs, the selection of the optimal concentration was performed using Infinicyt™ 2.0.5 software (Cytognos SL, Salamanca, Spain). The analysis of each concentration was conducted according to the hits number for IgM, IgG, and the match between both isotypes. After this grouping, only the top 5 mAbs were visualized for each isotype. This top 5 list was grouped according to the detected Ab isotype and the analysis described above (IgM, IgG, or both Abs for the same Ag).

For the generation of the databases and graphical representations, we used spreadsheets and Infinicyt™ 2.0.5 (Cytognos SL, Salamanca, Spain) based on information from the neXtProt [32], String 11.5 [33], Reactome [34], and Human Protein Atlas (www.proteinatlas.org, accessed on 19 June 2022) [35,36,37] databases.

## 3. Results

### 3.1. Quantitative Immunoglobulin Levels

Statistical analysis of the Ig concentration displayed a non-normal distribution independently of the compared subsets (diagnosis (MBL/CLL), disease evolution (c-CLL/p-CLL), therapy response (CLL-PFT/CLL-TFT), or IGHV mutational status (M-IGHV/U-IGHV)) except for IgG2 (Appendix A).

We observed significant variations in the IgA levels according to disease evolution (c-CLL vs. p-CLL) and for IgG1 according to IGHV mutational status (M-IGHV vs. U-IGHV) (Appendix A).

In a further analysis according the clinical/diagnosis groups, several patterns were observed: (i) IgG3 and IgA decreased according to disease evolution (c-CLL to p-CLL) (Appendix A); (ii) the Ig serum levels were constant in the MBL and c-CLL subsets, but a reduction in the Ig concentration was observed in the CLL-PFT subset, and slight increase in the Ig serum levels was observed in the CLL-TFT subset (Appendix A); and (iii) the mutational status was also reflected in reduced IgG1 serum levels when U-IGHV and M-IGHV were compared (Appendix A).

According to the disease evolution and progression, we observed the expected distribution across high-abundance Ig in the serum. A deep view within the IgG subclasses also revealed differences according the disease evolution and progression (Figure 1). For example, IgG1 was higher and IgG4 was lower when comparing MBL (Figure 1A), p-CLL (Figure 1B), and CLL-PFT (Figure 1C). Nevertheless, the IgG2 serum levels for the CLL (Figure 1A), c-CLL (Figure 1B), CLL-TFT (Figure 1C), and U-IGHV (Figure 1D) subsets were similar to those of IgG1; and for M-IGHV (Figure 1D), their serum levels were higher than for IgG1.

### 3.2. Autoantibodies to Known Tumoral and Neoplasm Antigen Proteins

The serological profile of aAb against tumor antigens (TAs) showed a high homogeneous distribution for IgG compared to IgM (Appendix A). Overall, the IgM profile displayed a normalized signal intensity that was 0.3 to 15 fold higher than the Ref.; meanwhile, the intensity was 0.2 to 42 fold higher than the Ref. for IgG (Appendix A).

In the top 10, the TAs observed for IgM were mostly against mutagenic proteins (ALDOA, ANXA1, THYG, BGLR, or LAMP-2), the most frequent subcellular localization of which was the cytoplasm (Cyt.) and vesicles (Ves.). In addition, IgM against CMV, protein E4 of human papillomavirus type 11 (HPV-11 E4), and CA72-4 was identified (Figure 2A and Appendix A). On the other hand, the 10 most frequent TAs detected by IgG were CA15-3 and CD274 (both are well-known tumoral biomarkers), the E6 protein HPV-11 (HPV-11 E6), and nonmutagenic proteins (PD-L1, NP1L3/NP1L4, MMP-2, Ox40L, and GBU4-5/TDRD-12), among others. Regarding subcellular localization for all of the detected TAs, a particular subcellular location was not observed (membrane (Mem.), Cyt., Ves., secreted (Secr.), and nucleus (Nuc.)) (Figure 2B and Appendix A).

Among these observations of the top 10 TAs, it is relevant to highlight that there were both isotypes against the E4 protein of HPV type 6a (HPV-6a E4) in 52% of the studied samples and 38% against SCCA (Figure 2C).

According to the disease progression and IGHV mutational status, the majority of the TAs in the top 5 list were proteins located on Ves. (AGTR1, LAMP-2, IL-6, MMP-2, Ox40L, CAP7, APOB, and TWEAK). Independently of the studied subset, LAMP-2 and AGTR1 were detected by IgG aAbs, while MMP-2 was detected by IgM aAbs except for CLL-TFT (Appendix A). In addition, ENOB was a TA observed by both isotypes and in all subsets. In this analyzed cohort, most of the detected TAs were nonmutagenic proteins and located in Ves. (AGTR1, MMP-2, Ox40L, and TWEAK).

### 3.3. Profile of Antimicrobial Antibodies according to Stimulation Persistent Antigenic Array (SPA Array)

Firstly, the SPA array displayed the best performance in the serum screening at 0.0001 µg/µL [mAg] (Table 2).

At 0.0001 µg/µL [mAg], the majority of mAgs with a higher normalized signal intensity than the Ref. (NIST standard) were detected by both Ig isotypes followed by IgM and IgG. Most of them were viruses that affected the respiratory and gastrointestinal tracts (Appendix A).

mAgs detected by IgM corresponded to bacteria and viruses responsible for infections of the respiratory and gastrointestinal tracts, mucosa, blood, and erythema (Appendix A); while mAgs detected by IgG corresponded to gastrointestinal viruses (Appendix A). The presence of both isotypes in the serum was detected against viruses, bacteria, and fungi related to respiratory, gastrointestinal, hepatic, mucosa, erythematous, and salivary gland infections (Appendix A).

The serological profile of mAbs against mAgs showed a high homogeneous distribution for IgM compared to IgG (Appendix A). Overall, the normalized signal intensity was 0.5 to 4 fold higher than the Ref. for IgM; meanwhile, it was 0.3 to 1 fold higher to the Ref. for IgG (Appendix A).

The 10 most frequent mAgs in the studied cohort are displayed in Appendix A. We observed that most of the IgM was against viruses and bacteria from respiratory tract infections (influenza A (FLU.Ohio)*,* human metapneumovirus (hMPV.16 and hMPV.9)*,* respiratory syncytial virus B (RSV.B)*,* SARS-CoV-2 and *Streptococcus pneumoniae* (SPN)), and a fungus from a mucosal infection (*Candida albicans* (CAV)) (Figure 3A); while it mostly detected IgG Abs against respiratory tract infection virus (FLU.Ohio, Measles (MV), RSV.B, and SARS-CoV-2) and bacteria (*Leptospira biflexa* (LB.Patoc1) and SPN) (Figure 3B). Only between 1.5% and 4.5% of patients in the cohort had both serum isotypes present against SPN, CMV, Echovirus (EV), Flu.Ohio, and RSV. B (Figure 3C).

According to the disease progression/treatment and IGHV mutational status, among the five most frequent mAgs there was the presence of IgM against hMPV.16 and IgG against EV and parvovirus (PV) regardless of the analyzed subset (except for the CLL-TFT group, in which hMPV.16 and EV were not detected) (Appendix A).

The majority of the top 5 mAgs were related to viral gastrointestinal infections (CMV, hepatitis C virus (HCV), hMPV.16, herpes simplex viruses 1 and 2 (HSV.1 and HSV.2), mumps virus (MuV), MV, rubella virus (RV), SARS-CoV-2, EV, PV, norovirus GI.1 and GII.4 (NoVGi.1 and NovGii.4), rotavirus (RV), astrovirus (HastV.1), and enterovirus (EV.COXB3)), a fungus (CAV), and a bacteria (*Salmonella typhi* (STY)). Among these observations, it is relevant to highlight those gastrointestinal viruses regardless of the subset in which they are detected due to the presence of IgG mAbs in the serum. For the CLL, c-CLL, and M-IGHV subsets, there was the presence of IgM mAbs against the hemolysin E protein of STY; while for p-CLL, CLL-TFT, and U-IGHV there was the presence of IgG mAbs against the outer membrane protein of STY.

## 4. Discussion

Over the years, the prognosis and therapies of CLL patients have been significantly improved. The leading causes of death in CLL patients are mainly due to comorbidities and infections because CLL patients have a dysregulation of the immune system that is both innate and adaptive, which increases the risk of infection. Moreover, MBL patients are much more susceptible to infection than the general population [20,21,38]. In the disease progression for most CLL patients, the number of nonclonal B cells is decreased and the number of B-CLLs is increased. This leads to a reduced level of Abs in the PB, thereby triggering hypogammaglobulinemia [11,20,21]. Hypogammaglobulinemia is found in most CLL patients upon diagnosis, while a minority (15%) continue to show normal blood Ig levels [39]. For all these reasons, we systematically evaluated the serum Ig levels in CLL and a profile of aAbs and mAbs that these patients present at several stages of the disease (from the early stage to the most advanced one).

In this studied cohort, we observed that the IgG and IgM showed similar serum levels regardless of the disease stage and IGHV mutational status. In the general population, the relative abundance of Ig isotypes in the serum was higher for IgG and IgA and lower for IgM and IgE [40,41]. IgM, which is the first isotype that is expressed during the development of B cells, is responsible for providing a rapid response (primary humoral immune response) and regulatory tissue homeostasis. IgM recognizes a wide variety of pathogenic molecules (i.e., nucleic acids, lipids, and proteins) that are phylogenetically conserved and have normal levels in serum of between 10^2.5^–10^4^ µg/mL [40,42,43,44,45,46,47]. On the other hand, the IgG isotype is an Ab that has a high affinity for its Ag because it occurs—like the rest of Ig isotypes other than IgM—during the evolution of Ab repertoires in response to antigenic stimulation and cytokine regulation. The normal serum levels that have been described were between 10^3^ – 10^4^ µg/mL [40,42,43,47,48]. In CLL, it is well characterized that there are a high number of anergic and active lymphocytes that trigger an ineffective response, thereby increasing the production and release of IgM [18] and equalizing their serum concentrations with respect to the IgG isotype. On the other hand, the fact that IgM average levels were increased in the samples of CLL-TFT ([IgM] > 10^2^ µg/mL) with respect to CLL-PFT ([IgM] < 10^2^ µg/mL) and MBL ([IgM] > 10^2^ µg/mL) was due to the type of therapy received (treatment with ibrutinib). As indicated by Forconi et al., during the period of time that patients are treated with ibrutinib, their B-CLLs are deprived of the stimuli present in the TME. This induces in the B-CLLs an increment in IgM on the surface and decreases the expression of the rest of the Ig isotypes [5].

Regarding the IgA levels, we observed a significant reduction according to the disease progression (c-CLL and p-CLL) and even when patients were receiving treatment (10^2^–10^1^ µg/mL). As Ishdorj et al. [38] and Vitale et al. [11] previously reported, the decrease in IgA levels was correlated with disease progression and development because its levels were not restored when the patients were in targeted therapy (ibrutinib) (IgA normal range is 10^2.5^–10^4^ µg/mL). Likewise, IgA serum levels were low in U-IGHV ([IgA] > 10^1.5^ µg/mL), which was associated with a poor prognosis [38]. This contrasted with the results observed by Peppas et al. [45] that several solid tumors exhibited increased IgA serum levels. Another relevant observation was that the average levels of IgM ([IgM] = 10^2.3^ µg/mL), IgG1 ([IgG1] = 10^2.9^ µg/mL), and IgG2 ([IgG2] = 10^2.7^ µg/mL) were increased in CLL-TFT with respect to the average levels in MBL ([IgM] = 10^2.2^ µg/mL, [IgG1] = 10^2.8^ µg/mL, and [IgG2] = 10^2.6^ µg/mL), although the levels before the disease development could not be restored (IgM and IgA: 10^2.5^–10^4^ µg/mL) [20,42].

Concerning the IgG subclasses in CLL, there are still many unknown aspects regarding the relative quantitative variations in serum. In this study, we detected higher levels of IgG2 in comparison with those of the IgG3 and IgG4 subclasses for the studied cohort, which was in accordance with previous studies by Copson et al. [49] and Lacombe et al. [50] but in contrast with one by Freeman et al. [20]. However, when the variations in the serum levels were determined for disease groups (MBL, p-CLL, and treatment), the known pattern of relative abundance in the serum was maintained [40,41,48]. These patterns are important and are related to the function of each IgG subclass: IgG1 mainly recognizes viruses and soluble and membrane protein Ags [41]; IgG2 recognizes soluble cytokines and bacterial polysaccharides and inhibits receptor–ligand interaction [41,43]; IgG3 recognizes viruses and Ags present on the membranes of blood cells [41]; and IgG4 is produced by repeated and long-term exposure to Ags in noninfectious environments (decreasing Ab-dependent cellular cytotoxicity and complement-dependent cytotoxicity prevents the binding of other isotypes of Ig to C1q) and recognizes helminths and parasites [17,41]. Therefore, imbalance levels in IgG subclasses could generate susceptibility and/or the severity associated with the disease [41,48]. This bias was observed for the CLL, c-CLL, CLL-TFT, and U-IGHV groups for IgG1 and IgG2, which had similar serum quantitative levels. Therefore, CLL, c-CLL, CLL-TFT, and U-IGHV presented reductions in the immune response against viruses, cell recruitment (monocytes, neutrophils, eosinophils, dendritic cells, and macrophages) and isotype changes. In the case of M-IGHV, this bias occurred for IgG2, so this group will not trigger a cytotoxic response via complement activation, it had low cell recruitment and as consequence, it could be more susceptible to infections by viruses, helminths, and parasites due to reduced levels of the rest of the IgG subclasses.

As is known, the immunogenicity of any biomolecule is due to its intrinsic properties (i.e., biochemical and structural properties), subcellular and cellular locations, mutations, and cross-reactive Abs (because the length of the epitope recognized by the Ab varies between 4 and 12 amino acids), among other characteristics [51,52,53]. In autoimmune diseases, aAgs are common intracellular molecules that B cells (via BCR recognition) and Abs cannot access under normal conditions [54]. In this study, the cohort presented mostly IgM Abs against aAgs located in Cyt. and Ves. When this same observation was made with respect to the diagnosis, disease evolution, treatment, and IGHV mutational status, there was an increased presence of IgM Abs against aAgs located in Ves. This was consistent with the expected and reported results for autoimmune diseases. In addition, as the disease became more aggressive (p-CLL, CLL-PFT, CLL-TFT, or U-IGHV), more de novo aAgs were detected.

In general, TAs are cellular products that occur in both tumoral and normal cells; however, the levels of these products in blood and other body fluids are increased when a tumor develops. They include a wide variety of biomolecules (Ags on the surface of cells, Cyt. proteins, enzymes, hormones, oncofetal Ags, and oncogenes, among others) that are routinely used in clinical cancer diagnoses to evaluate the diagnostic and prognostic situation of the patient, thereby allowing an association of the serum levels with the stage of the tumor [55,56,57].

In this analyzed cohort, there was the presence of IgG against the tumoral markers CA 15-3 (soluble MUC1) and CD274 (soluble PD-L1); while for CA 72-4 (TAG-72), there was the presence in the serum of the IgM isotype. This might suggest that CA 72-4 is one of TAs related to disease evolution; meanwhile, CA 15-3 and CD274 were already present at constant levels prior at diagnosis. This contrasted with what was observed for CA 15-3 in other types of cancers such as breast cancer as well as a low expression of PD-L1 in the B-CLLs [58,59,60]. On the other hand, SCCA is a protein located in the Cyt. and Nuc. that is widely expressed in certain organs (such as the esophagus or female reproductive organs) under normal physiological conditions [61]. For these reasons and based on the results obtained from the studied cohort, it is feasible that 50% of the patients had only IgG. Moreover, the presence of both isotypes in 39% of the cases could be due to de novo Abs production against other protein epitopes because SCCA is a mutagenic protein. However, when the analysis was carried out in relation to the patient subsets, we observed that for the MBL patients, most of them displayed IgM against these TAs, which may have been indicative of cross-reactivity and consequently of de novo production of Abs.

It seems relevant to highlight that the list of top 5 TAs included IgM against MMP-2 and IgG against AGTR1 and LAMP-2, both of which were isotypes against ENOB regardless of the analyzed subset.

MMP-2, a protein involved in metabolic and enzymatic processes located in the Ves., is responsible for the degradation of the extracellular matrix due to its enzymatic action on type IV collagen. Its levels increase as a tumor develops; which causes a local decrease in pH in the microenvironment that triggers the activation of proteolytic enzymatic processes (the degradation of type IV collagen), thereby helping in the separation of tumor cells from the tissue and promoting their expansion [62,63]. Therefore, the presence of the IgM isotype in clinical subsets indicates an increase in the enzymatic activity of the degradation of the extracellular matrix (which allows the dissemination of tumor cells).

On the other hand, AGTR1, which is a component of the renin–angiotensin system (which is responsible for the maintenance of homeostasis), has highly detectable expression levels in the central nervous system, adrenal glands, thyroid, digestive system, or urinary system, among others, under normal physiological conditions [64]. Likewise, LAMP-2 is responsible for stabilizing the structure of the lysosome (an acid catabolic organelle located in all mammalian cell types except mature erythrocytes) by forming part of the lysosome membrane and functioning within lysosomal trafficking, exocytosis, chaperone-mediated autophagy, and cholesterol fusion and transport [65]. Therefore, the presence of IgG aAb and not IgM aAb for both TAs may have been due to the degradation/inhibition of the activity of these proteins in the TME. All of this contrasted with the observations in other cancers such as colon (for LAMP-2) or breast (for AGTR1), which presented an increase in expression levels and consequently a decrease in MMP-2 levels [64,65].

Likewise, in the current study, the presence of both isotypes of Ig against ENOB indicated that this might be a constitutive level in the early stages of disease because it is a protein involved in the catabolism and anabolism of glucose [66] and is increased with disease progression and evolution; this was demonstrated by Griggi et al. [67] in a study conducted in CLL patients. Apart from this, ENOB acts as a plasminogen receptor that facilitates pathogen intrusion due to the degradation of the extracellular matrix (as was observed with *S. pneumoniae* infections, which was one of most prevalent mAgs in the studied cohort) [66].

Bearing in mind that IgM commonly are polyclonal, polyreactive, and low-affinity Abs, it seems that TAs recognized by IgM are correlated with tumoral progression and disease evolution as Ags increase their relative abundance (independently of mutagenesis and alterations). Therefore, it is expected that these aAbs will be generated de novo because IgM is the first Ig produced after the activation of B cells [46]. Likewise, the IgG isotype is an Ab secreted by memory B lymphocytes and plasma cells [48]. It is known that in aging, IgG aAbs increase, thereby revealing one of the immune tolerance mechanisms [51]. In autoimmune diseases, IgG and IgM increase, which is associated with disease pathogenesis [44]. Therefore, IgG aAbs might be generated in early stages of the disease and produced during progression due to the activation and differentiation of memory B lymphocytes into the plasma.

Dysregulation of the immune system is a key feature in the early stages of CLL and in disease progression due to an impaired humoral response and a weakened cellular response [12]. As discussed above, hypogammaglobulinemia is related to a functional deficiency in T lymphocytes and non-clonal B cells. It is correlated with the duration and the disease stage and also is associated with a susceptibility to infections [68,69]. It was previously observed that CLL patients with hypogammaglobulinemia were more prone to recurrent bacterial infections. In particular, low levels of IgG were associated with infections by *Streptococcus* and *Haemophilus*. They also had a delayed hypersensitivity to *C. albicans*, diphtheria toxin, and the Paramyxoviridae family [69,70]. In this cohort, we observed that 12% of the patients presented IgG Abs against *S. pneumoniae* vs. 9% of patients who presented IgM. In the case of *C. albicans*, ~12% of the patients presented IgG. When the comparison was performed for disease progression, among the five most recurrent mAgs, we observed in all subsets the presence of both isotypes of Ig against *C. albicans* except for CLL-PFT (IgM) and CLL-TFT (IgG). This was expected due to the hypersensitivity of these patients to fungal infections caused by *C. albicans* and recurrent reinfections due to *S. pneumoniae*.

Another aspect to consider was the intrinsic characteristics of the disease evolution and therapies received. The more advanced stage of tumor development and more lines of treatment that have been received; more prone a patient is to suffering infections. Therefore, in the clinic it is necessary to identify the mAg (bacteria, virus, or fungus) and the severity of the infection (mild, moderate, or severe) [21,24,25,68]. Likewise, monitoring via serological tests of certain viral infections is recommended before starting any treatment (such as the hepatitis B and C viruses, CMV, human immunodeficiency virus (HIV)*,* HSV, or varicella-zoster, among others) [25].

It was previously observed that patients treated with chlorambucil (an alkylating agent) were more likely to present bacterial infections associated with *S. aureus*, *S*. *pneumoniae*, *H. influenzae,* or Gram-enteric bacteria. In these patients, the focus of the infection is mostly on the respiratory tract and mucous membranes. In the treatment with fludarabine (a purine analogue), patients are not only more likely to suffer from the aforementioned infections, but also from opportunistic (*Listeria*, *Mycobacteria* or *Pneumocystis*) and viral (varicella-zoster and HSV) infections. The use of novel therapies is not exempted from the infection risk. Treatment with ibrutinib or venetoclax makes subjects more likely to suffer from severe pneumonia. On the other hand, therapy with ibrutinib also contributes to infections that originate in the respiratory and urinary tracts, opportunistic infections by *Pneumocystis jirovecii,* and an increased risk during the first 6 months of treatment of fungal infections (*Aspergillus* and *Cryptococcus*); while therapy with venetoclax makes them more susceptible to respiratory and opportunistic infections due to *Aspergillus*, *P*. *jirovecii*, varicella-zoster or *C*. *albicans*, among others. Apart from making patients more prone to opportunistic and HSV infections, treatment with alemtuzumab also produces a CMV reactivation [68,69,70].

In the studied cohort, most of the 10 most common mAgs were respiratory tract viruses. This cohort presented both isotypes of Ig against HPV-6a E4 (which is responsible for amplifying the genome, suppressing cell differentiation, and allowing virion release). This highlighted the ability of this virus to reinfect such patients. When this observation was made on the five most frequent mAgs according to the biological group, most were gastrointestinal viruses. In all groups, there was the presence of IgG Abs against EV and PV, which suggested that these types of virus generated immunity before the development of the disease, while infection by hMV.16 occurred after its development.

Finally, it should be added that in the study cohort, the presence of both isotypes of Ig against CMV occurred in similar proportions (~11% for IgG and ~9% for IgM). When observed according to biological groups, patients belonging to groups with a worse prognosis (p-CLL and CLL-TFT) presented Ig G, while the stages of a better prognosis (MBL and c-CLL) presented Ig M. It should be noted that for the IGHV mutational status with a poor prognosis (U-IGHV), both isotypes of Ig were present compared to M-IGHV (in which the Ig M isotype was detected); therefore suggesting that this subset of patients had a high probability of virus reactivation.

## 5. Conclusions

The characterization of Ig serum levels (such as Ig A) in CLL is relevant to the diagnosis and the disease evolution and/or progression. In CLL, it was generally observed that Ig serum levels were reduced as the disease progressed. On the other hand, this study confirmed that CLL patients had recurrent infections by respiratory tract viruses and that Ig M levels were increased against these TAs, which suggested a stochastic state between de novo aAbs and immune tolerance (aAgs related to metabolism intracellular pathways). In addition, differential profiles in Ig isotypes seemed to provide relevant information about the most optimal therapeutical treatment. In summary, the dynamics of Ig quantitative levels and Ag profiling could be highly useful in patient stratification and prognostic factors for disease evolution.

## Figures and Tables

**Figure 1 cancers-15-00891-f001:**
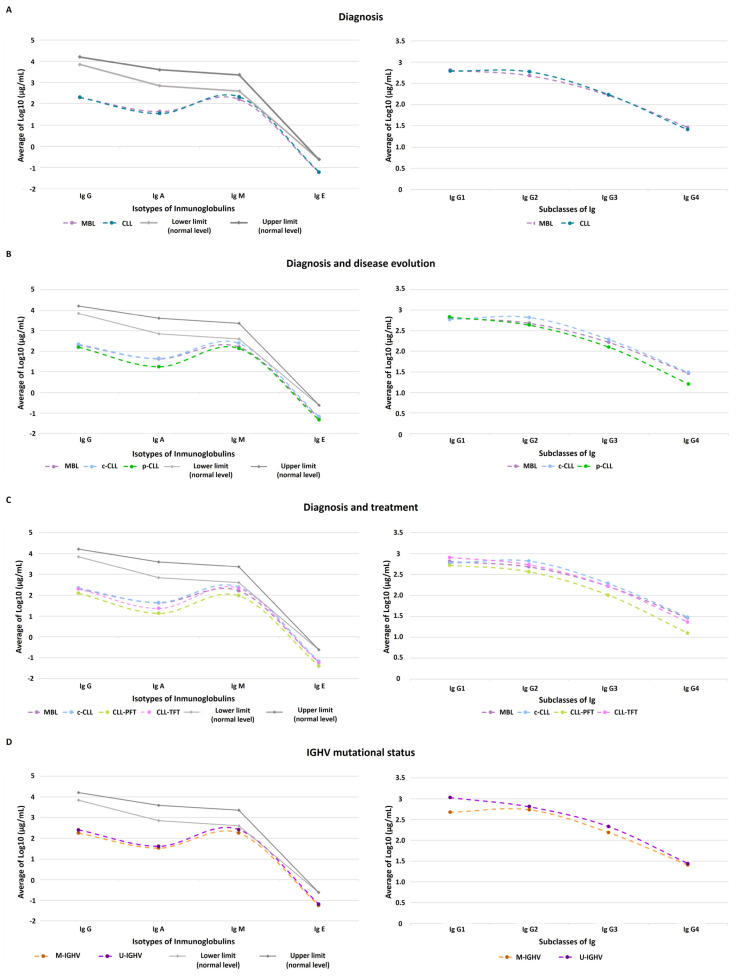
Quantitative immunoglobulin profiles according to clinical/diagnosis subsets: (**A**) MBL and CLL groups; (**B**) MBL, c-CLL, and p-CLL groups; (**C**) MBL, c-CLL, and treatment (CLL-PFT and CLL-TFT) groups; (**D**) IGHV mutational status (M-IGHV or U-IGHV).

**Figure 2 cancers-15-00891-f002:**
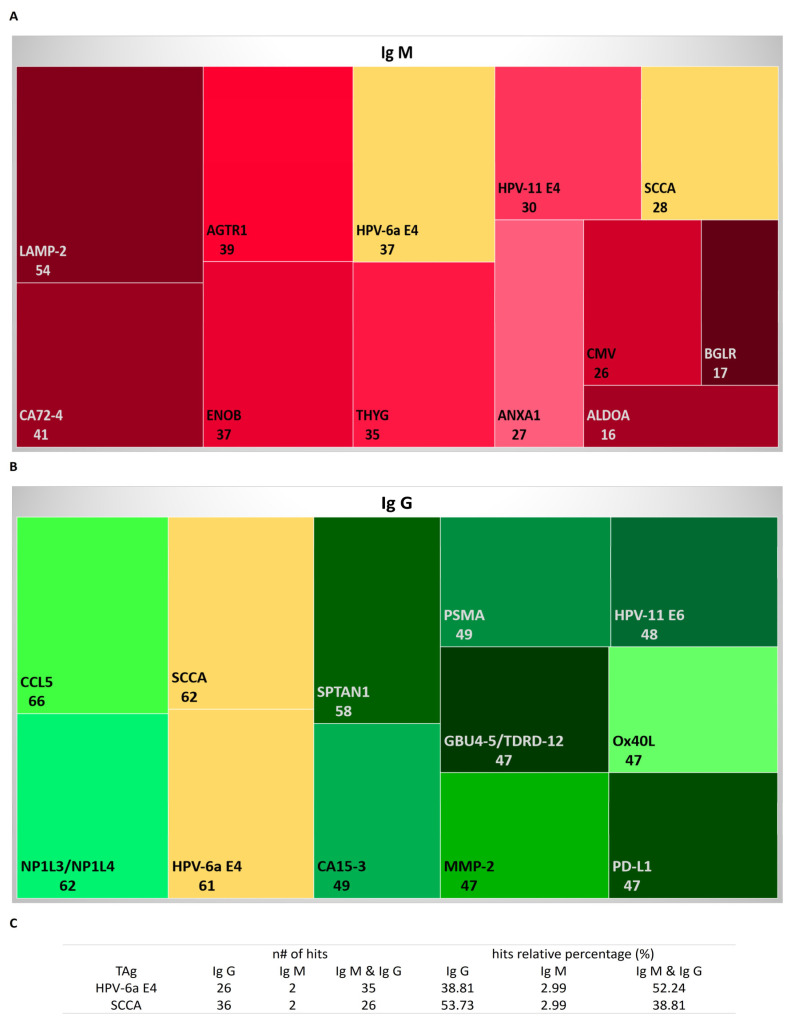
Quantitative summary of top 10 tumoral antigens (TAs) detected in this study: (**A**) TAs detected by IgG autoantibodies (aAbs) in the studied cohort; (**B**) TAs detected by IgM aAbs in the studied cohort; (**C**) number (*n*#) and percentage (%) of samples for each detected aAb with respect to the total studied cohort.

**Figure 3 cancers-15-00891-f003:**
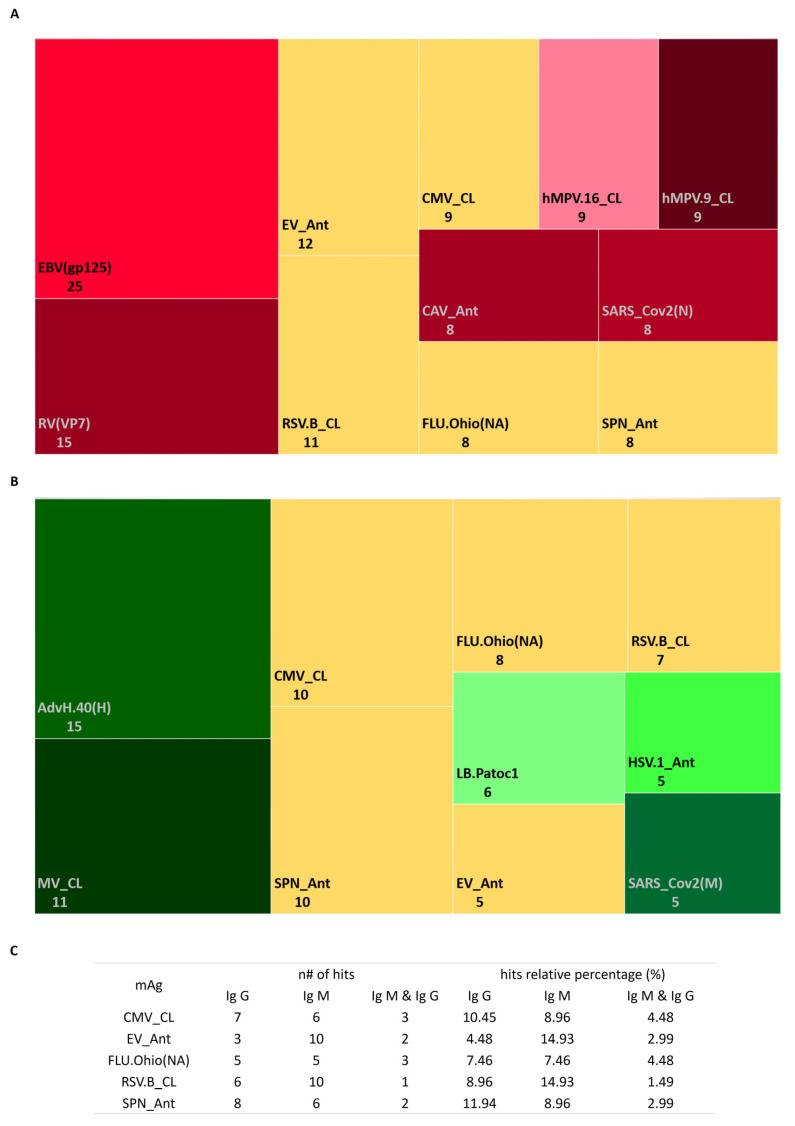
Quantitative summary of top 10 microbial antigens (mAgs) detected in this study: (**A**) mAgs detected by IgG antibodies (Abs) in the studied cohort; (**B**) mAgs detected by IgM Abs in the studied cohort; (**C**) number (*n*#) and percentage (%) of samples for each detected mAg with respect to the total studied cohort.

**Table 1 cancers-15-00891-t001:** Clinical characteristics of patient cohort.

Clinical Information		Frequency (*n*)	Percentage (%)
Gender	Female	29	43.3
	Male	38	56.7
Age	≤65	24	35.8
	>65	43	64.2
Diagnosis	MBL	10	14.9
	CLL	57	85.1
CLL status	Stable/constant	42	73.7
	Progression	15	26.3
Binet stage	A	54	80.6
	B	6	9
	C	7	10.4
Rai stage	0	45	67.2
	I	5	7.5
	II	10	14.9
	III	1	1.5
	IV	6	9
Treatment status	Previous to first-line	9	60
	Time from first-line	5	33.3
IGHV gene status	Mutated	45	67.2
	Unmutated	22	32.8
Cytogenetic	Normality	24	35.8
	Abnormality	43	64.2
Total		67	100

**Table 2 cancers-15-00891-t002:** Global number of microbial antigens (mAgs) detected in this screening after analysis of 37 different mAgs.

	mAgs						
Concentration (µg/µL)	S < Ref.	Ref. > S					
	n# of mAgs	IgM	IgG	IgM and IgG	IgM	IgG	IgM and IgG
0.08 and 0.06	11	11	7	8	3	0	3
0.01	3	26	1	7	7	0	3
0.001	8	16	1	12	4	0	5
0.0001	12	10	3	12	1	0	5
0.00001	5	12	1	19	5	0	5
		mAgs (n#)			Top 5 mAgs (n#)		

Note: mAgs, microbial antigens; *n*#, number; S, normalized signal intensity; Ref.: NIST standard.

## Data Availability

Data sets are available at BIODATA Repository USAL (https://biodata.usal.es/share.cgi?ssid=ad631b3d918e4031bd129589a11bc2f5) upon request to corresponding author.

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
