# Peer review of "Systematic Evaluation of Antigenic Stimulation in Chronic Lymphocytic Leukemia: Humoral Immunity as Biomarkers for Disease Evolution"

_cancers, 2023, doi:10.3390/cancers15030891_

Round 1

Reviewer 1 Report

The manuscript covers an important topic and the findings are of interest. However the manuscript should need to be further inproved

Major comments

1) Authors should present the data reagarding the presence of autoantibodies against tumor antigens and antibodies against infectious antigens in a more clear and comprehensive manner

2) Authors need to use a control group consisted of healthy individuals in order to compare the findings from patients with CLL with the the control group. Is there any difference betweeen healthy donors and patients with CLL?

3) Authors should clarify in detail, how the findings from their analysis explain the autoimmune phenomena and the susceptibility to infections observed in patients with CLL

4) The methodology should be presented in a more clear way

5) Figures are of low quality and it is very difficult for the reader to see the details 

6) Authors use abbreviations without clear explanations in the text

Author Response

Thanks to the reveiwer for all your insighful comments and suggestions which have been taken into consideration in this updated version of the manuscript. In fact, a new analysis has been included with healthy serum samples from the National Institute of Standards ( NIST) from USA. In addition, the methodology section has been improved and the full manuscript has been updated. The figures have been edited and increase the quality. Moreover, the manuscript has been double-checked and revised in order to make it more readable and avoid missunderstanding with several abbreviations. 

Reviewer 2 Report

The way to investigate the diagnosis, pathogenesis and prognosis of CLL is original and very interesting compared to the classic prognostic indications. This method is argued with well-studied sophisticated methods and extensive translational references to the pathogenesis of the disease.

However, there is no specific reference to the aforementioned method, in the context of the different therapies present in the literature. To the evolution, prognosis and resistance of the disease to therapies based on the study method and also to infections in reference to therapies (only on page 13 from 593 to 595), with the proposed study method. Furthermore, the aforementioned method of prognostic classification is complex in its realization and the possible practical advantage compared to the current prognostic methods, in the daily clinical routine, is not well explained.

I therefore believe it is necessary, in addition to proposing a new method of investigation of the disease, also to demonstrate its prognostic advantages, compared to the already existing ones or at least the clinical-therapeutic advantages in proposing new treatments (anti-infectious?) or discriminate new methods of choice with respect to existing ones.

Author Response

Thanks to the reviewer for all your insightful comments and suggestions which have been taken into consideration in this updated and revised version of the manuscript. Hence, the prognostic advantages have been discussed in this updated version of the manuscript. Moreover, several paragraphs have been included regarding the methology; among more references about the methodology, in order to provide a more described methodology. 

Round 2

Reviewer 1 Report

I propose to reduce the length of Introduction

Otherwise, i consider the response to my comments is adequate